

# Modelling of shallow water table dynamics using conceptual and physically based integrated surface water-groundwater hydrologic models

Mohammad Bizhanimanzar[1], Robert Leconte[1], Mathieu Nuth[1]

[1]Department of Civil Engineering, Université de Sherbrooke, Sherbrooke, J1K2R1, Canada

*Correspondence to*: Mohammad Bizhanimanzar (Mohammad.Bizhanimanzar@usherbrooke.ca)

**Abstract.** We present a new conceptual scheme of the interaction between unsaturated and saturated zones of the MOBIDIC (Modello Bilancio Idrologico DIstributo e Continuo) hydrological model which is applicable to shallow water table conditions. First, a hydrostatic equilibrium moisture profile was assumed for simulating changes in water table levels. This resulted in a

water table based expression of specific yield, which was included in the coupled MOBIDIC-MODFLOW modelling framework for capturing shallow water tables fluctuations. Second, the groundwater recharge was defined using a power type equation based on infiltration rate, soil moisture deficit and a calibration parameter linked to initial water table level, soil type and rainfall intensity. Using the Water Table Fluctuation (WTF) method, the water table rise for a homogeneous soil column under a pulse of rain with different intensities (up to 30 mm/day) the parameter of

the proposed groundwater recharge equation was determined for four soil types i.e., sand, loamy sand, sandy loam and loam. The simulated water table levels were compared against those simulated by MIKE-SHE, a physically based integrated hydrological modelling system simulating surface and groundwater flow. Two numerical experiments were carried out: a two-dimensional case of a hypothetical watershed in a vertical plane (constant slope) under a 1cm/day uniform rainfall rate, and a quasi-real three dimensional watershed under one month of

measured daily rainfall hyetograph. The comparative analysis confirmed that the simplified approach can mimic simple and complex groundwater systems with an acceptable level of accuracy. In addition, the computational efficiency of the proposed approach (MIKE-SHE took 180 times longer to solve the 3D case than the MOBIDIC-MODFLOW framework) demonstrates its applicability to real catchment case studies.

## 1 Introduction

Over the last decades, a number of integrated surface-subsurface hydrologic models were developed. The main objective of such models is to conceptualize the hydrologic cycle in an integrated way, particularly by coupling the surface and subsurface (unsaturated and saturated zones) hydrological processes. Such integration is particularly important in humid regions where the water table is close to the surface and runoff generation is dominated by variable source area mechanisms (Dunne and Black, 1970; McDonnell and Taylor, 1987). In this runoff generation mechanism, infiltrated water enters the water



table, which rises until it reaches the surface, often in valley bottoms, creating areas where any additional precipitation results in saturation excess runoff (McDonnell and Taylor, 1987). Investigation of such runoff mechanism at the catchment scale can be addressed using physically based models in which the unsaturated zone (UZ) and the saturated zone (SZ) are either explicitly or implicitly coupled. In an explicit coupling approach, a 1D Richards' equation for the unsaturated zone is coupled to a three-

dimensional saturated flow. This approach assumes that flow in the unsaturated zone is only vertical and the groundwater recharge is computed using an iterative water table correction process. MIKE-SHE (Refsgaard and Storm, 1995) is an example of such coupling approach. In the implicit coupling of the unsaturated-saturated zones, the whole subsurface flow process is described using a 3D variably saturated flow equation without an explicit distinction in the interaction between unsaturated and saturated zones (Camporese et al., 2010; Kollet and Maxwell, 2006). This approach more truly reflects the physical

processes governing flow but is computationally more expensive as compared to explicit approaches, which themselves require considerable computer resources to solve the unsaturated flow equation at the watershed scale. There is a third group of integrated surface-subsurface hydrologic models in which already existing hydrologic and groundwater models are coupled, such as SWAT-MODFLOW (Chung et al., 2010); TOPNET-MODFLOW (Guzha and Hardy, 2010); GSFLOW (Markstrom et al., 2008). The challenging issue regarding the applicability of these externally coupled models in humid shallow

water table regions is related to the inconsistencies in the conceptualization of the interaction between unsaturated and saturated zones. Seibert et al. (2003) distinguishes three types of interactions between unsaturated and saturated zones based on water table levels:

Type 1. The water table is relatively deep and there is only one-directional interaction between UZ and SZ. This means the soil moisture state in the unsaturated soil is independent of the groundwater level and the role of groundwater in runoff

generation process is not considered.

Type 2. The water table is about at the root zone level. The unsaturated soil can get water from the capillary rise in groundwater so the interaction becomes two-directional as groundwater recharge can be either positive or negative to create hydrostatic equilibrium with the water table. The unsaturated soil profile is assumed to have a constant vertical extension (unsaturated moisture capacity (the maximum amount of moisture the unsaturated layer can hold) is constant during the course of

simulations). Such assumptions were made by different hydrological models such as TOPMODEL (Beven et al., 1995), SWAT (Neitsch et al., 2011) and MOBIDIC (Castillo et al., 2015).

Type 3. The water table is very close to the surface and unsaturated moisture capacity can no longer be assumed to be constant as the water table fluctuates. This is the case when water table rise results in a decrease in the unsaturated capacity and a small amount of the infiltration causes a significant rise in groundwater level (due to the significantly small moisture deficit in

unsaturated zone). This is the main thrust of this paper as we introduce a conceptual approach compatible with the existing MOBIDIC framework for applications where strong interactions between the UZ and SZ exist. More specifically, the calculation of groundwater recharge is revisited as the existing linear function in MOBIDIC (groundwater recharge is a linear function of moisture storage in gravity reservoir) is incompatible with the nonlinear behaviour of moisture fluxes between UZ



and SZ typical of shallow water table regions. Such nonlinear behavior is associated with nonlinear decrease in magnitude of specific yields as the shallow water table rises to the soil surface.

In this paper, we applied the water table fluctuation method (WTF) (Healy and Cook, 2002) to investigate the behavior of groundwater recharge in shallow water tables (up to 1.5m deep) for four soil types i.e., sand, loamy sand, sandy loam and loam in a soil column under a single pulse of rainfall with different intensities. These simulations were carried out using MIKE-SHE as a reference model. MIKE-SHE assumes a constant specific yield (in saturated flow computation process) and the computed water table is corrected based on a specified mass balance threshold error in UZ-SZ coupling process. The simulated water table levels of MIKE-SHE are then considered as the 'true' hydrologic response of the system and are utilized to reformulate the groundwater recharge model component of MOBIDIC. The accuracy of the proposed changes is first tested in a two-dimensional case where subsurface water is simulated in a vertical plane with constant slope. A constant rainfall rate is applied and the rise in groundwater levels is affected by groundwater recharge and by the lateral interaction between the saturated computational grids. In a second numerical experiment, the accuracy of the approach is further evaluated at the catchment scale and under unsteady rainfall where the simulated water table levels of the two models (MIKE-SHE as the reference model and MOBIDIC-MODFLOW) are compared.

## 2 Water table fluctuation method

The Water Table Fluctuation (WTF) method is a simplified approach for the determination of groundwater recharge of an unconfined aquifer based on groundwater level fluctuations. This method is based on the assumption that the rise in groundwater levels is due to the groundwater recharge (Healy and Cook, 2002). Considering the groundwater budget for a representative element, (Fig. 1), any change in the water table level (groundwater storage) would be due to a combination of recharge to groundwater $(R)$, inflow from upstream cell $(Q_u)$, outflow to the downstream cell $(Q_d)$ and evapotranspiration from groundwater $(ET_{GW})$ as follows:

$$\Delta S_{GW} = R + Q_u - Q_d - ET_{GW} \tag{1}$$

Where $\Delta S_{GW} [LT^{-1}]$ is the change in groundwater storage. Assuming that water table rise is solely due to the recharge of groundwater requires the sum of other fluxes in Eq.1 to be zero. This means that the determination of the groundwater recharge using WTF is best applicable over short periods (hours to days) after onset of rainfall (before any significant redistribution of groundwater recharge to the other fluxes) (Healy and Cook, 2002). Therefore:

$$R = S_y \frac{\Delta h}{\Delta t} \tag{2}$$

Where $S_y[-]$ is the specific yield and $\frac{\Delta h}{\Delta t}$ is the change in water table over $\Delta t$. Application of the WTF method requires an estimation of the specific yield which is defined as the volume of drained water per unit drop in water table and aquifer area (Nachabe, 2002):





$$S_y = \frac{V_w}{A.\Delta h} \qquad (3)$$

Where $V_w[L^3]$ is the volume of drained water, $A[L^2]$ is the area of the aquifer and $\Delta h[L]$ is the change in water table level. The specific yield is also defined as the difference between water contents at saturation $\theta_{sat}$ and at field capacity $\theta_{fld}$ (the moisture

level below which water cannot be drained by gravity (Nachabe, 2002)). Such constant value for specific yield, however, holds only for deep water tables where changes of the soil moisture profile in the unsaturated zone due to water table drop are relatively small and the volume of drained water can be approximated as $(\theta_{sat} - \theta_{fld}) \times \Delta h$ (Fig. 2a). In shallow water tables however, the specific yield is small (in Fig. 2b, shaded area is smaller than $(\theta_{sat} - \theta_{fld}) \times \triangle h$ ) and approaches zero when the capillary fringe zone extends up to the soil surface (Nachabe, 2002).

**3 Models description**

**3.1 MIKE-SHE**

MIKE-SHE is one of the widely used physically based integrated surface-subsurface hydrological model for a wide range of spatiotemporal applications ranging from detailed theoretical (single soil column) to operational watershed scale studies (Graham and Butts, 2005). It has a modular structure for computation of the hydrological processes with different levels of

complexity, which is advantageous particularly in large-scale watershed studies (Kollet et al., 2017). A detailed description of the computation of the hydrological processes in MIKE-SHE can be found in Storm (1991) and DHI (2014). We present only the computation of flow in the unsaturated and saturated zones and their coupling approach.

**3.1.1 Unsatuarted flow**

The unsaturated flow is described using the one dimensional Richards' equation as (Downer and Ogden, 2004):

$$\frac{\partial \theta}{\partial t} = \frac{\partial}{\partial z}\left(k(\theta)\frac{\partial \psi}{\partial z}\right) + \frac{\partial k(\theta)}{\partial z} - S(z) \qquad (4)$$

where $\theta[-]$ is the volumetric water content, $k(\theta)[LT^{-1}]$ is the unsaturated hydraulic conductivity, $\psi[L]$ is the pressure head in unsaturated soil, $z[L]$ is the elevation in vertical direction, and $S[T^{-1}]$ is a sink term (e.g. root extraction). The numerical solution of the Richards' equation is based on the implicit finite difference method in which the soil layer is discretized into the computational nodes and the discretized equation is solved with prescribed upper (rainfall rate or ponded water) and lower

(water table level) boundary conditions (DHI, 2014). The main challenge with solving the Richards' equation is the computational burden, as it requires fine discretizations in terms of space and time (time steps of seconds to minute). This can be problematic particularly in long term watershed scale studies.





### 3.1.2 Saturated flow

Saturated flow in MIKE-SHE is computed using the three-dimensional saturated flow equation as:

$$\frac{\partial}{\partial x}\left(K_x \frac{\partial h}{\partial x}\right) + \frac{\partial}{\partial y}\left(K_y \frac{\partial h}{\partial y}\right) + \frac{\partial}{\partial z}\left(K_z \frac{\partial h}{\partial z}\right) - Q = S \frac{\partial h}{\partial t} \tag{5}$$

Where $K_x, K_y, K_z \ [LT^{-1}]$ are the saturated hydraulic conductivity along the $x, y$ and $z$ axes, respectively, $h \ [L]$ is the

groundwater head, $Q \ [T^{-1}]$ is the source/sink term and $S$ is the storativity coefficient $[-]$. The storativity coefficient is either the specific yield, $S_y$, for unconfined aquifers or the specific storage , $S_s$, for confined ones (DHI, 2014). Equation (5) is solved numerically over computational grid squares using the finite difference method with the Preconditioned- Conjugate Gradient (PCG) solver, which is also included in MODFLOW (DHI, 2014).

### 3.1.3 Unsaturated-Saturated zone coupling

The explicit coupling approach implemented in MIKESHE has the advantage of employing different times steps for each zones (seconds to minutes for UZ and hours to day for SZ), which makes the system computationally less expensive compared to the implicit coupling approach (DHI, 2014). However, employing different time steps for the UZ and SZ may result in the generation of mass balance errors in calculating the water flux between the two zones due to 1) an incorrect value of $S_y$ (recommended value is $\theta_{sa} - \theta_{fld}$) in computing the saturated flow and 2) a fixed water table level while the UZ progresses to

the next time step (Storm, 1991). The coupling between UZ and SZ follows an iterative process by which the water table is adjusted until the accumulated mass balance error for the entire soil layer (UZ+SZ) falls below a prescribed threshold. The process also calls for adjusting the soil moisture profile.  The groundwater recharge is calculated once the iterative process has converged.  It should be noted that $S_y$ is considered as constant in the computational process.  The algorithm of iterative adjustments of the water table can be summarized as follows (Storm, 1991):

Step 1) at the beginning of the UZ time step, the accumulated mass balance error for the entire soil layer (UZ+SZ) of a soil column ($E_{cum}$) is calculated.

Step 2) if the accumulated error falls below the acceptable level, the water table does not require any adjustments and groundwater recharge is calculated as follows:

$$R = \frac{\partial}{\partial t} \int_{z_h}^{z_g} \theta(z) dz + \sum q_u \tag{6}$$

Where $R$ is groundwater recharge, $\sum q_u$ is the net inflow to the unsaturated soil. The calculated recharge is applied as the upper boundary condition of the SZ module and simulation advances to the next time step.

Step3) if the calculated error is beyond the acceptable threshold, then the water table adjustment is required. Depending on the sign of $E_{cum}$ the water table will be either raised or lowered.

      3-a) if $E_{cum} < 0$, it means less water is stored in the profile and therefore water table has to be raised. Using the

30         updated moisture profile in UZ (only the last three nodes of the UZ profile (above water table) are updated), go to step 1 and repeat.



3-b) if $E_{cum} > 0$, it means more water is stored in the profile and therefore the water table is lowered. Using the updated moisture profile in UZ (same as described above only for the three lowest nodes of the UZ profile), go to step 1 and repeat.

The iterative water table adjustment continues until the calculated $E_{cum}$ falls below the acceptable limit. Then the new groundwater recharge is calculated as:

$$R = -\frac{(h^* - h)S_y}{\Delta t} \tag{7}$$

Where $h^*$ and $h$ are the water table after and prior the adjustments, respectively. The calculated groundwater recharge will then be applied as the upper boundary condition of the SZ module and simulation advances to the next time step.

## 3.2 MOBIDIC

Modello Bilancio Idrologico DIstributo e Continuo (MOBIDIC) (Castelli et al.,2009) is a distributed continuous hydrologic model in which the components of the hydrologic system are conceptualized as a system of inter-connected reservoirs. Such a conceptual formulation of the model makes it computationally more efficient especially in large scale watershed modeling (Castillo et al., 2015). The detailed description of the model is available in Castelli et al. (2009) and Castillo et al. (2015).

### 3.2.1 Unsaturated flow

In MOBIDIC, the unsaturated zone is described by two interconnected reservoirs i.e., the gravity and capillary reservoirs, to account for corresponding gravity and capillary forces in the unsaturated soil. The water content at field capacity ($\theta_{fld}$) is the threshold below which moisture is entirely held in the capillary reservoir. The gravity reservoir interacts with the saturated zone via percolation to the groundwater and it may also redistribute the additional moisture to the downstream cells (lateral flow in the unsaturated zone) (Castillo et al., 2015). The capillary reservoir serves moisture to the plant roots (or evaporation if there is no plant in the computational grid) and can also take moisture from lower groundwater via capillary rise (Castillo et al., 2015). The moisture capacities of the gravity reservoir and the capillary reservoir, $W_{g_{max}}$ [L] and $W_{c_{max}}$[L], are defined as (Castillo et al., 2015):

$$W_{g_{max}} = d.(\theta_{sat} - \theta_{fld}) \tag{8}$$
$$W_{c_{max}} = d.(\theta_{fld} - \theta_{res}) \tag{9}$$

where $d[L]$ is the unsaturated soil thickness and $\theta_{sat}[-], \theta_{fld}[-], \theta_{res}[-]$ are the water content at saturation, field capacity and residual, respectively, which are determined based on soil texture classification (Rawls et al., 1982). At each time step, water available for infiltration (net precipitation plus the ponded water on the soil surface) is determined. The infiltration rate takes the minimum value among the saturated hydraulic conductivity, the allowable moisture capacity in gravity reservoir and the available water. The infiltration rate, however, might be underestimated for low permeable soils in dry conditions when the capillary force at the soil surface is not negligible Castelli (1996). The gravity





reservoir is replenished by the infiltrated water. Absorption flux (moisture that is extracted by from the gravity reservoir to the capillary reservoir) is calculated as (Castillo et al., 2015):

$$Q_{as} = \min\left\{W_g + I, \kappa\left(1 - \frac{W_c}{W_{c,max}}\right)\right\} \tag{10}$$

Where $\kappa[T^{-1}](0 \leq \kappa \leq 1)$ is a linear coefficient that controls the rate of the moisture transfer between the two reservoirs.

Finer soils typically have higher $\kappa$ value because of low downward moisture gradient. The groundwater recharge ($Q_{per}$) is calculated as (Castillo et al., 2015):

$$W_{g,u} = W_g + I - Q_{as} \tag{11}$$

$$Q_{per} = \begin{cases} \min\left\{\gamma\ W_{g,u}, \frac{\left[W_{g,u} + \left(\frac{z_w}{d} - 1\right)W_{g,max}\right]}{dt}\right\} & if\ z_w\ \geq 0 \\ \min\left\{\frac{(W_{g,max} - z_w - W_{g,u})}{2dt}, (W_{g,max} - W_{g,u})/dt\right\} & if\ z_w\ < 0 \end{cases} \tag{12}$$

Where $z_w[L]$ is the depth to water table, $W_{g,u}[L]$ is updated moisture in gravity reservoir and $\gamma[T^{-1}](0 \leq \gamma \leq 1)$ is a

coefficient that controls the rate of groundwater recharge. The available moisture storage in the gravity reservoir can also contribute to lateral flux to the adjacent cell and is calculated as (Castillo et al., 2015):

$$Q_{lat} = \beta(W_{g,u} - Q_{per}) \tag{13}$$

Where $\beta[T^{-1}](0 \leq \beta \leq 1)$ is a coefficient that determines the rate of lateral flow and $W_{g,u}$ is the updated moisture state in gravity reservoir after reduction of groundwater recharge. It should be noted that, as the unsaturated flow is strictly vertical in

MIKESHE, we ignored the lateral redistribution of moisture in gravity reservoir ($\beta$=0) to make the structure of the two models consistent.

### 3.2.2 Saturated flow

Saturated flow in MOBIDIC is described either by a simplified linear reservoir (conceptual scheme) or by MODFLOW (Harbaugh et al.,2000)(Three-dimensional Finite-Difference based groundwater model). We used the latter to facilitate

comparison        between        the        simulated        results        of        MOBIDIC        and MIKE-SHE regarding their respective conceptualization of the saturated flow and its effect on the interaction between UZ and SZ.

### 3.2.3 Coupling Unsaturated and Saturated zones

Unlike MIKE-SHE, coupling UZ-SZ in MOBIDIC is not based on an iterative water table correction procedure. Based on the

calculated water table level in the previous time step, the recharge to groundwater can be positive (recharge to groundwater) or negative (extraction from groundwater) Castillo (2014). The latter occurs when the saturated storage is bigger than moisture storage in the gravity reservoir ($W_g$). Therefore, the water table in the subsequent time step falls to establish a hydrostatic equilibrium        with        moisture        level        in        gravity        reservoir



Castillo (2014). However in the former, groundwater is recharged by a higher moisture level in the gravity reservoir. This results in a rise in water table in the subsequent time step Castillo (2014). During the dry periods, the capillary reservoir may also receive water from the capillary rise from water table which is calculated as (Castillo et al., 2015):

$$Q_{cap} = \frac{\left[\left(\frac{d_w}{\psi_1}\right)^{-n} - \left(\frac{\psi}{\psi_1}\right)^{-n}\right]K_s}{1 + \left(\frac{\psi}{\psi_1}\right)^{-n} + (n-1)\left(\frac{d_w}{\psi_1}\right)^{-n}} \tag{14}$$

where $d_w[L]$ is the mean distance of the unsaturated layer to water table, $\psi_1[L]$ is the bubbling pressure, $n[-]$ is the product of pore size distribution index (Rawls et al., 1982) and pore size disconnected index (Brooks and Corey, 1964). The unsaturated soil water pressure ($\psi$) is a function of saturation state of the layer and pore size distribution index ($m$) and is calculated as (Castillo et al., 2015):

$$\psi = \psi_1 S^{-1/m} \tag{15}$$

$$S = \frac{W_c + W_g}{W_{g_{max}} + W_{c_{max}}} \tag{16}$$

It should be noted that the capillary rise is computed where the water table is within the soil profile ($z_w \leq d$), otherwise the capillary rise flux will be zero.

## 4 Water Table Fluctuation method for a soil column using MIKE-SHE

We use the WTF method over a soil column to understand how the rise in groundwater is affected by rainfall intensity, soil types and depth to water table level and how much of the infiltrated rain will percolate to the groundwater. Using a single soil column with closed boundaries on the sides and bottom, the rise in groundwater level will only due to the recharge (see eqs. 1 and 2). In shallow water table regions, as the water table rises the specific yield decreases nonlinearly and its value approaches zero when capillary fringe is extended up to the soil surface. This means that, in such conditions a small amount of rain may result in a significant rise in water table (up to the soil surface if the water table is close enough to the surface such that the capillary fringe reaches the surface) as investigated by (Abdul and Gillham, 1984, 1989; Buttle and Sami, 1992; Sklash and Farvolden, 1979; Waswa and Lorentz, 2015). The schematic illustration of the method is given in Fig. 5. If the infiltrated water is completely transferred to the groundwater, the rise in water table will be:

$$\Delta h = \frac{I}{(\theta_s - \theta_{fld})} \tag{17}$$

Where $I[L/T]$ is the infiltrated water and $\Delta h[L]$ is the rise in water table level. $\Delta h$ is referred as the 'reference rise'. However, depending on the initial water table level and rainfall intensity, it is possible that the water table rise be smaller than $\Delta h$ (e.g. for relatively deeper water table and/or low rainfall rate) or that the groundwater recharge takes a value greater than the infiltration rate. The 'actual rise' of the water table then can be calculated same as in Eq.2. Therefore, by defining $\left(\frac{Re}{I}\right)$ we can evaluate how the rise of the water table is related to the precipitation rate (assuming no infiltration excess runoff) and depth to the water table. If $\frac{Re}{I} > 1, =, < 1$ means that the actual rise of the water table is larger, equal, or smaller than the reference rise,




respectively. Once again, in MIKESHE, the $S_y$ is a constant input of the saturated module and the changes regarding its value (decrease in its value when depth to water table decreases) is being captured by the coupling procedure explained in section 3.1.3.

The procedure was repeated for different soil types i.e., sand, loamy sand, sandy loam and loam. The hydraulic properties of
the soils are given in Table 1. This was done to further investigate the significance of the rise in the water table of soil types with different characteristics in response to a given pulse of rain. In shallow water tables, loamy soils with larger capillary fringe range have substantially smaller specific yield values which result in a higher rise of water table compared to the sandy soils Gillham (1984). Therefore, its actual rise in the water table is expected to be larger than the reference rise. However, as loamy soils have smaller hydraulic conductivities compared to sand, their water table response will be delayed especially for
deep water tables. To avoid this delayed response, we focused our analysis to cases where the initial depths to the water table are at a maximum of 1.5m. Computational time steps were one second and one minute for UZ and SZ, respectively to avoid numerical instabilities in the simulated water tables by MIKESHE.

## 4.1 Simulation results

Simulation results of the water table rises are shown in Fig. 4. Each dotted curve in the plots is associated with a specific initial
depth to water table (ranging from 0.3 m to 1.5 m) and rainfall rate. The plots in Fig.4 are divided into two zones i.e., $\frac{Re}{I}>1$ (water table rise exceeds the reference rise, red dots) and $\frac{Re}{I}<1$ (water table rise is less than the reference rise, blue dots). Whereas the lower bound of the chosen initial depth to water table in Fig. 4 is 1.5m, the upper bound changes from 0.3 m for sand to 0.9 m for loamy soil (see Table 1). This is due to the differences in water retention characteristics of the soils and initial moisture deficit in unsaturated zone. For example, in sandy soil with 30 cm initial depth to water table (capillary fringe is
15.98 cm), the moisture deficit is 14.98 mm. So, for the infiltration rate bigger than this value, the unsaturated soil becomes completely saturated and water table rises up to the soil surface as it can be seen in Fig. 4 in which the upper curve ends at a rainfall rate of about 15 mm/day. The same argument to the other soil types. For loamy soil with capillary fringe value of 40.12 cm, the initial depth to water table is 90cm corresponds to the moisture deficit 23.97 mm in unsaturated zone. Therefore, the upper curve for loamy soil ends at this depth to water table and rainfall intensity.

For a given initial water table level, moving from low to high rainfall intensities results in increasing the ratio of recharge to infiltration, shifting from a situation where $\frac{Re}{I}<1$ (blue dots) to $\frac{Re}{I}>1$ (red dots). Also, while the case where $\frac{Re}{I}<1$ is more common in sandy soils, the opposite is found as soil texture becomes finer (e.g. loamy soil). This means that the actual water table rise in fine textured soils (loamy sand to loam) is almost always larger than would be expected based on using a constant value for specific yield $= (\theta_s - \theta_{fld})$.

Note that for all soil types analyzed, the $\frac{Re}{I} > 1$ values show small sensitivity to increases in rainfall intensity as the depth to water table gets larger, e.g. in sandy soils and loamy soils when the water table is deeper than 50cm and 1 m, respectively.



This can be attributed to the existence of significant initial soil moisture deficits found at larger initial water table depths. On the other hand, as the water table gets closer to the soil surface, an increase in the rainfall rate generates a significant water table rise, yielding to a substantial increase in $\frac{Re}{I}$ . This demonstrates the importance of the soil moisture deficit (and its ratio to the precipitated rainfall) in water table rise magnitude and dynamics.

## 5 Changes in conceptualization of the UZ-SZ interactions in MOBIDIC-MODFLOW

As discussed in section 3, MOBIDIC's unsaturated soil depth ($d$ in Eq.(7) and (8)) remains constant during the course of a simulation (both in rainy and subsequent draining periods). This is due to its conceptualization of UZ and SZ as their interaction is not aimed to address the reverse relationship between them in very shallow water tale cases. The proposed changes are intended to make the model applicable for such cases. The changes are as follows:

1-It is assumed that the unsaturated soil layer thickness, $d$, is no longer a constant input of the model and changes with water table fluctuations. Hence, the total moisture capacity of the reservoirs ($W_{g_{max}}$ and $W_{c_{max}}$) are determined similarly as in Eq. (7) and (8), but with replacing $d$ by the depth to water table. Therefore, water table rise/fall results in a decrease/increase in moisture capacities of the reservoirs. Such assumption is valid when groundwater level is treated as a moving boundary and there is a continuous transfer of moisture between the unsaturated and saturated zones.

2-The specific yield in calculation of groundwater head by MODFLOW is determined based on a soil water retention model (e.g. Brooks and Corey, 1964) and hydrostatic equilibrium assumption in unsaturated zone (suction profile in unsaturated zone changes from steady state to another over the changes in water table) (Hilberts et al., 2005) Duke (1972):

$$S_y = (\theta_s - \theta_r)\left\{1 - \left(\frac{\psi_1}{z_w}\right)^m\right\} \qquad z_w > \psi_1 \tag{18}$$

Where $z_w$ is depth to water table and other variables are as defined above. Validity of hydrostatic equilibrium assumption is justified in shallow water table regions where redistribution of the infiltrated water and an equilibrium moisture profile occur immediately Bierkens (1998). Therefore, with any changes in water table level, specific yield is adjusted for the calculation of groundwater dynamics in the next time step. This is especially important in shallow water table cases as specific yield decreases significantly as water table gets closer to the soil surface.

3- The groundwater recharge $R[LT^{-1}]$ is a function of available moisture in the gravity reservoir, unsaturated soil moisture deficit and infiltration rate and is calculated as:

$$R = I \left(\frac{W_g}{1+(Deficit)}\right)^{-\omega} \tag{19}$$

$$Deficit = \left(W_{g_{max}} + W_{c_{max}}\right) - (W_g + W_c) \tag{20}$$



Where $I[LT^{-1}]$ is the infiltration rate, and $\omega[-]$ is a function of soil type, infiltration rate and water table level, which determines how much water will be transferred to the saturated zone. It should be noted that only gravity reservoir contributes to the groundwater recharge (similarly as for the original conceptualization of groundwater recharge in MOBIDIC).

The proposed conceptualization of the interaction between UZ and SZ are shown in Fig. 5. Starting with a hydrostatic equilibrium moisture profile in the UZ, the available moisture in the reservoirs, $W_g$ and $W_c$ as well as the moisture deficit of in the unsaturated profile are determined. The rise in water table results in a reduction of the moisture capacities of the reservoirs $W_{g_{max}}$ and $W_{c_{max}}$, as well as $W_g$ and $W_c$ and the moisture deficit in unsaturated profile. When water table falls, however, there is an increase in these quantities. It should be noted that, unlike MIKESHE, the unsaturated and saturated modules in MOBIDIC-MODFLOW run with a daily time step which means any change in the water table level causes an instantaneous adjustment of unsaturated moisture profile.

## 5.1 Interpretation of the new groundwater recharge equation in MOBIDIC-MODFLOW

To interpret the proposed groundwater recharge equation (Eq. (19)) let us first analyze its limit values. As $\frac{W_g}{1+(Deficit)} < 1$, a decrease in $\omega$ results in a decrease in $R$ (and vice versa). The coefficient $\omega$ can be either positive or negative. When positive, recharge to the groundwater reservoir is larger than the infiltration rate and when negative, groundwater recharge is a portion of the infiltration. The specific case for which $\omega = 0$ occurs when R = I. Such flexibility in value range of $\omega$ is important as it makes the model applicable for cases where adding small rainfall amount can cause a significant water table rise, which may occur in humid regions. However, its precise determination can be problematic as it is a complex function of water table level, soil type, and rainfall (infiltration) intensity. In general, for a known water table level, an increase in rainfall rate will require a larger $\omega$ values to match the water table rise of the MIKESHE and the MOBIDIC-MODFLOW models. Equation 18 was calibrated for $\omega$ by using simulated water table rises of MIKESHE. This allowed investigating how combinations of soil types, rainfall intensities and water table depths influence the behavior of groundwater recharge and how this can be captured using the conceptual UZ-SZ formulation presented here.

## 5.2 Simulation results

Figure 6 displays the values of parameter $\omega$ in Eq. (19) obtained by fitting the water table rises of the MOBIDIC-MODFLOW model to the water table rises simulated by MIKESHE for the cases presented in Fig. 4. The negative and positive values of $\omega$ are shown in red and blue dots, respectively. It can be seen that for a specified water table level, increasing the rainfall rate requires higher values of $\omega$ to fit the water table rise of MOBIDIC-MODFLOW to that of MIKESHE. Furthermore, the relationship is nonlinear. This further shows the complexity of the interactions between the UZ and SZ under shallow water table conditions. Note that in sandy soils most of the $\omega$ values are negative, while the opposite is found for the fine textured soils such as loam. This is because the water table rise for loamy soils are larger than for sandy soils (smaller specific yield and available moisture deficit in unsaturated profile). Comparing the scatterplots in Fig. 6, we realized that unlike loamy soils,





in sandy soils with relatively deeper initial water tables, the changes in $\omega$ (with changes in rainfall rates) is quite small. This is related to the initial moisture deficit in unsaturated profile as discussed in section 4.1.

These $\omega$ values are arranged in a look-up table that can be used to simulate the UZ-SZ interactions in MOBIDIC-MODFLOW in the following way: at each time step, based on the specified water table depth, soil type and rainfall rate a value of $\omega$ is obtained by interpolation from the look-up tables. The determined $\omega$ are then used for computing groundwater recharge using Eq. (18). The recharge is next applied as the upper boundary condition to the layers. Using the specific yield calculated with Eq. (18), MODFLOW updates the groundwater head and water table level values. The updated levels are then transferred to MOBIDIC-MODFLOW for the determination of moisture storages in the capillary and gravity reservoirs ($W_g$ and $W_c$). This procedure is repeated for the entire simulation period. The next section compares the performance of the MOBIDIC-MODFLOW model against MIKESHE in two cases, a hypothetical vertical 2-D slope model and a small a pseudo-hypothetical watershed.

## 6 Application of the MOBIDIC-MODFLOW scheme in two test cases

The appropriateness of the proposed changes in the conceptualization of UZ-SZ interactions in shallow water tables are tested by comparing its predicted water table results against those of MIKESHE in two test cases. The description of the test cases follows.

### 6.1 Two-dimensional case

So far, our analysis focused on a single soil column in which the coefficient of the proposed groundwater recharge, $\omega$ (Fig. 6), were determined based on the simulated water table rise by MIKESHE (Fig. 4). In order to analyze the performance of the proposed approach in cases where there is lateral interaction between saturated computational grids, simulations of the water table rise were performed for a small soil box measuring 14m long, 1.5m high and 1m wide filled by sandy soil identical to the one used in the soil column analysis. The soil box has closed boundary conditions on $x = 0$, $x = 14$m and bottom side and it is assumed that initial water table is $h(x, 0) = 0.2$m. Since the rainfall is continuous during the entire simulation period, the configuration of the box results in higher expected water table rise in the toe of the slope (shallower water table) and generation of saturation excess runoff starting from downhill moving to the uphill. The hillslope is discretized into 28 columns ($\triangle x = 0.5$m) and the vertical discretization of the unsaturated layer in MIKESHE is $\triangle z = 1$cm. A 10 mm/day constant rainfall is applied for 20 days and the simulated water table rises of the two models are compared.

### 6.1.1 Simulation results

The simulated water tables of the two models are shown in Fig. 7. The two models in general show similar behavior in predicted water tables. The slight mismatch between the predicted water table heads of the two models can be attributed to the fact that in MOBIDIC-MODFLOW there is an immediate response of the groundwater to the precipitated water, that is, within the





same time step (one day). This is not the case in MIKESHE where the simulation time step is much shorter at one second. The saturated length (the length where water table at the soil surface) predicted by the two models are closely matched. This means simplifications in UZ-SZ interaction of MOBIDIC-MODFLOW can mimic the complex dynamical interaction between the two zones. The generated saturation excess runoff by two models were removed from the soil surface as the flow routing

module was not included in the simulations.

## 6.2 Catchment scale

### 6.2.1 Borden catchment

In order to further evaluate the suitability of the proposed conceptualization of UZ-SZ in MOBIDIC-MODFLOW, the water table fluctuations of the two models at the Borden catchment for the month of May 2015 were compared. The Borden

catchment is located approximately 70 km of Northwest of Toronto, Ontario (Jones et al., 2006). The site is about 20m wide and 80m long and it was subjected to several experimental (Abdul and Gillham, 1989) and numerical rainfall-runoff studies (VanderKwaak and Loague, 2001; Jones, Sudicky, and McLaren, 2008; Kollet et al., 2017) It is assumed that the catchment consists of a single homogeneous sandy soil identical as the one used in the previous simulations (Table 1) without any vegetation cover. The motivation for simplifying the watershed physiographic characteristics here was to evaluate the

MOBIDIC-MODFLOW in a 'real' watershed, while emphasizing on differences in UZ-SZ dynamics simulated by the conceptual approach as compared to a physically based numerical model.

The digital elevation model (DEM) of the Borden catchment has a spatial resolution of 0.5m and is available at www.hpsc-terrsys.de and is shown in Fig. 8a. The initial water table level was assumed to be at 1m below the catchment's outlet (at elevation 1.98m). The site has closed boundaries on the sides and it has a horizontal bedrock at elevation 0 m. The measured

rainfall at the Borden site for May 2015 used in the numerical experiment (data available at Environment Canada, 2015). The rainfall hyetograph (Fig. 10) consist of 10 events with total amount of 77.2 mm and maximum of 24.8 mm at day 25 where a significant rise in water table is expected. The catchment was subdivided into two zones based on the initial depth to water table as shown in Fig. 8b. Zone 1 represents the initial depth to water table less than 1.5m (red zone in Fig. 8 b) and cells in zone 2 have the initial depth to water table larger than 1.5m (blue zone in Fig. 8 b). The calculation of groundwater recharge

and interaction between UZ-SZ followed the modifications presented in section 5 (the $\omega$ parameter in Eq. (19) was made variable for water table levels less than 1.5m) and changes in groundwater head in zone 2 was simulated using the original conceptualization of MOBIDIC explained in section 3.2. Therefore, these two structures together cover the dynamics of the water table for the whole catchment. Note that, for water table depths greater than 1.5m, the moisture capacities of the reservoirs ($W_{g_{max}}, W_{c_{max}}$) remains constant with a rise or fall of the water table (section 3.2). Such distinction based on depth to the water

table was made to limit the extrapolation errors associated with the determination of $\omega$ for water table depths larger than 1.5m. Besides, for these zones the assumption of an inverse relationship between the unsaturated and saturated moisture storages is questionable as discussed in the introduction.



### 6.2.2 Simulation results

The difference in simulated water table level of the two models is shown in Fig. 9 for the selected simulation days after the rainy events. The predicted groundwater head of the models at the outlet of the catchment are also compared in Fig. 10. It can be seen that the two models generally compare well.

However MOBIDIC-MODFLOW is slightly predicting a higher water table rise as compared to MIKESHE following rainfall events (e.g. events on days 11, 18 and 25). This is attributed to the interaction of saturated flow between the two zones (the saturated flow moves from grids in zone 2 to zone 1). The magnitude of the overestimation decreases over the course of simulations (for example the differences in water table levels are smaller at day 23 than day 6). This further shows that how the interaction between the zones with different characteristic of the interaction between the UZ and SZ (zone 1 with inverse

relation between the unsaturated and saturated storage and zone 2 with direct relationship between them as classified in the introduction) can affect the overall behavior of the water table fluctuations at catchment scale. Therefore, that the rise in water table in a computational grid is also the result of the net incoming of saturated flow from surrounding grids for both models. That is why the predicted water able in MOBIDIC-MODFLOW is still slightly rising between two rainy days (from day 12 to day 17 and day 19 to day 24) as shown in Fig. 10. Consequently, the simulated water tables of the two models gradually

converge in the days following a rainfall event (for example between the days 14 to 17 and 19 to 24) as the transient behavior simulated by MIKESHE dissipates following a long redistribution period.

### 7 Discussion

Comparing the simulated water table levels of the two models at Borden catchment (Fig. 9 and 10) attests for the soundness of the proposed UZ-SZ interaction scheme of MOBIDIC-MODFLOW for shallow water tables. The Mean Absolute Error of

the predicted water table at the catchment outlet compared to MIKESHE is 0.013 m. As the water table rises in, the water table could be much shallower than expected by the ultimate value for specific yield ($\theta_s - \theta_{fld}$). So, the inclusion of a water table dependent expression for the specific yield in MOBIDIC-MODFLOW with the proposed groundwater recharge (Eq. (19)) would be able to capture its significant rise. In addition, the proposed groundwater recharge in MOBIDIC-MODFLOW has the capability to take into account the complex relation between rainfall intensity, soil, water table depth for the calculation of

the water table fluctuations. The parameter of the groundwater recharge ($\omega$ in Eq. (19)) was tuned to fit the water table rises simulated by MOBIDIC-MODFLOW to those of MIKESHE.

Moreover, as it was outlined in the introduction, the interaction between unsaturated-saturated zones in shallow water table cases is inverse as a rise in water table levels decreases the unsaturated moisture storage and vice versa. The modelling of such cases were mostly addressed using physically based numerical models in which a one dimensional Richards' equation is

coupled to a two-dimensional saturated flow. However the application of these complex models at the catchment scale can be cumbersome due to the required data and computational burden to run such models. Investigation of such cases using conceptual hydrologic models have not received much attention in the literature (e.g. Seibert et al., 2003). (Seibert et al., 2003)



conceptualized the inverse relation between UZ-SZ moisture storage of shallow water tables by dividing the hillslope into reservoir namely upslope and riparian. Saturated flow moves from upslope reservoir to the riparian and eventually to the adjacent stream. The moisture storage in unsaturated and saturated zones of each box is being determined by the water table level. Unlike in MOBIDIC-MODFLOW, the saturated flow in their model is computed based on the conceptual linear reservoir

approach which limits its applicability in representation of aquifer's heterogeneous properties and their effects on water table dynamics. Moreover, the coefficient of the groundwater recharge in their approach has not been explicitly formulated to capture the effect of soil type, rainfall rates and water table depths which are unavoidable in shallow water table regions.

Another advantage of the coupled MOBIDIC-MODFLOW presented in this paper lies in its capability to investigate phenomena such as saturation excess runoff without expensive computational burden. The physically based models such as

MIKESHE require very fine discretization of the unsaturated soil domain and require short time steps to avoid numerical instabilities. However, the proposed approach in MOBIDIC-MODFLOW works with identical time steps in both unsaturated and saturated zones and without discretization of the soil layer, which greatly improves the computational process. For example the execution time for the Borden case took 10 minutes using a PC Core i7 with 8GB of RAM for MOBIDIC-MODFLOW, while taking 30 hours with MIKESHE. This clearly indicates why such alternative modelling approaches are attractive. While

the ω parameter of the proposed groundwater recharge in Eq.(19) was determined for sand, loamy sand, sandy loam and loam, the model was tested for sandy soils where it was assumed that the unsaturated soil instantaneously adjusts to changes in water table level as discussed by (Bierkens, 1998). For finer textured soils however, the assumption of instantaneous equilibrium state between unsaturated and saturated zones may be questionable. The approach presented here should therefore be fully tested over a variety of soil textures to further confirm its exactitude to broader watershed conditions. The proposed changes

were intended to keep the MOBIDIC's number of calibration parameters low which is advantageous in model calibration as it does not raise the concern on equifinality (Beven, 2001). The coupling approach implemented in MOBIDIC-MODFLOW was tested in two-dimensional (single slope) and three dimensional cases (Borden catchment), both consisting of a homogeneous sandy soil. In addition to testing the proposed approach over a variety of soil textures, a next step will consist in extending the applicability of the approach against observations in real catchments where other hydrological processes, such as summer-fall

evapotranspiration and spring snowmelt, affect groundwater recharge in shallow aquifers.

## 8 Conclusion

In this paper we presented modifications to the conceptualization of the unsaturated-saturated zones in MOBIDIC for application in shallow water tables in which there is an inverse relationship between the unsaturated and saturated moisture storages.           The           key           variables           in           modelling           of           such           cases           are

1) groundwater recharge and 2) specific yield. Groundwater recharge is usually considered as a function of unsaturated moisture state and is a fraction of infiltrated moisture into the unsaturated soil layer. However in shallow water tables its value can be larger than infiltration due to the capillary fringe above the water table, which causes a quick and significant water table





rise (Jayatilaka and Gillham,1996). Investigation of such cases are mostly addressed using physical based models (e.g. Cloke et al., 2006) or using fully conceptual approaches (e.g. Seibert et al., 2003). We presented MOBIDIC-MODFLOW as a conceptual-numerical model in which changes in water table are assumed to establish an equilibrium moisture profile in the unsaturated soil zone. Employing the analytical expression of specific yield derived by Duke (1972) in the coupled model, its

dynamical behavior particularly in shallow water table regions was captured. The groundwater recharge in the model was defined with a power type equation whose parameter (ω) was determined using the Water Table Fluctuation method (WTF) along with the physically-based MIKESHE model for a homogeneous soil column with different initial water table levels under rainfall intensities up to 30 mm/day. The appropriateness of the proposed changes were evaluated by comparing the simulated water table behavior of the model against those of MIKESHE in a two-dimensional (single slope) case under

continuous uniform rainfall and a three-dimensional (Borden catchment) case under variable rainfall intensity. In the light of simulation results the following conclusions can be derived:

1- Inclusion of a dynamic specific yield in investigation of water table behavior in shallow water table is important. Indeed, when the water table is close to the soil surface, the significance of rise is much greater than would be expected based on the ultimate value of specific yield e.g., $\theta_s - \theta_{fld}$, as investigated by Jayatilaka and Gillham (1996). While

in MIKESHE this is accounted for by using the iterative water table correction method, in MOBIDIC-MODFLOW this is addressed by assuming a hydrostatic equilibrium state between UZ-SZ. To the authors' knowledge, this has not been taken into account in simplified coupling of a hydrologic and a groundwater model.

2- The analysis of the water table rise with different rainfall intensity, soil type and water table level shows a complex relation between the groundwater recharge and these factors. The simulation over different ranges of rainfall rates

and water table levels showed that fine textured soils with large capillary fringe (consequently small unsaturated soil moisture deficit) could have water table at soil surface even with a low rainfall intensity. This is important for the investigation of saturation excess runoff in lowland near river zones of the catchments.

3- The comparison of the simulated water tables of the two models at Borden case (Mean Absolute Error equals 1.3 cm) along with their execution times (30 hours with MIKESHE against 10 minutes with MOBIDIC-MODFLOW) clearly

demonstrates the applicability of simplified approach implemented in MOBIDIC-MODFLOW in investigation of the groundwater-surface water interaction. Further improvements of the approach with inclusion of other hydrological processes such as the effect of evapotranspiration on parameterization of the (ω) in future work will enable its applicability in real, practical applications.

**Acknowledgements**

This research was partly funded by a Discovery Grant from the National Science and Engineering Research Council of Canada and by the research center on water of the Université de Sherbrooke (GREAUS).



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


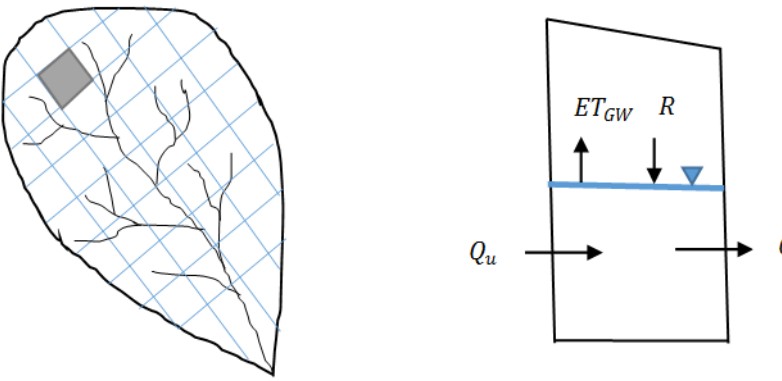

**Figure 1. Schematic view of computational grids in a catchment and corresponding input and output fluxes over the saturated zone.**

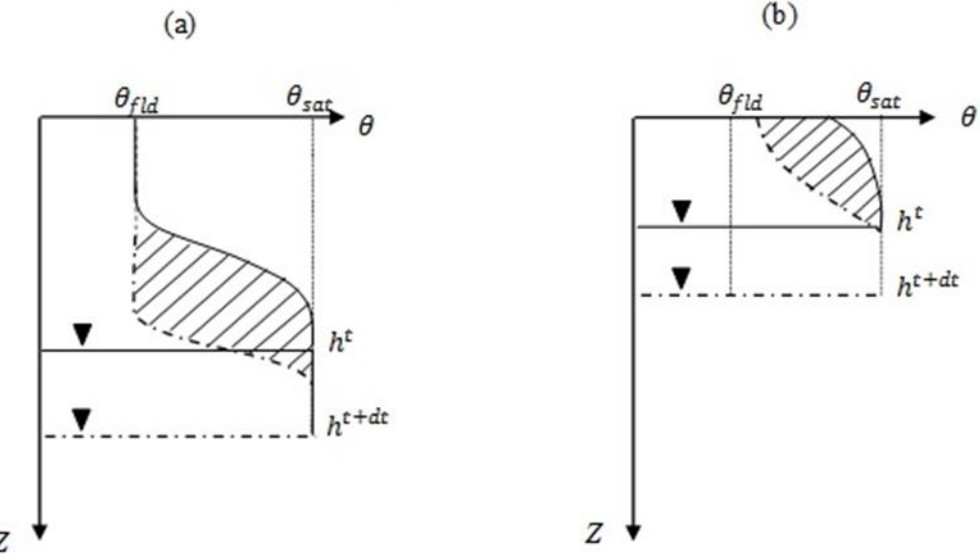

**Figure 2. Hypothetical soil moisture profile for (a) deep and (b) shallow water tables. The solid and dashed lines are the corresponding profiles before and after water table drops, respectively. The shaded area is the drained water due to a drop in the water table. Modified after Healy and Cook (2002).**



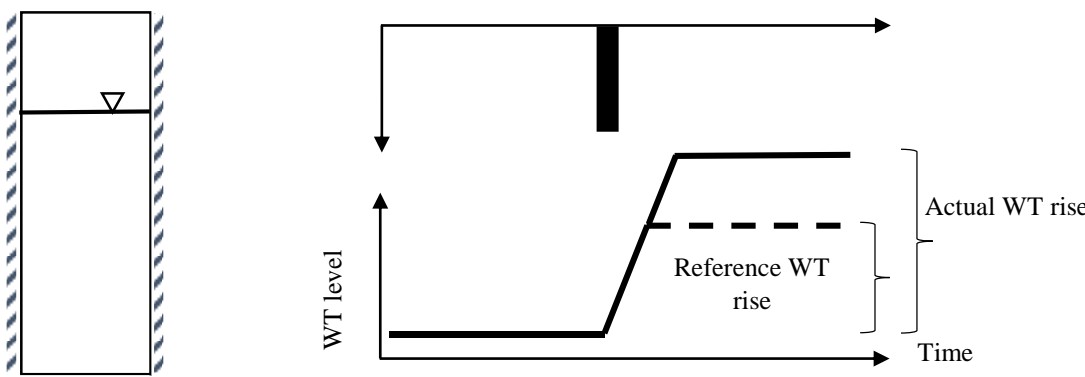

**Figure 3. Schematic representation of actual and reference water table rise using the WTF method for a soil column subjected to a single pulse of rainfall.**

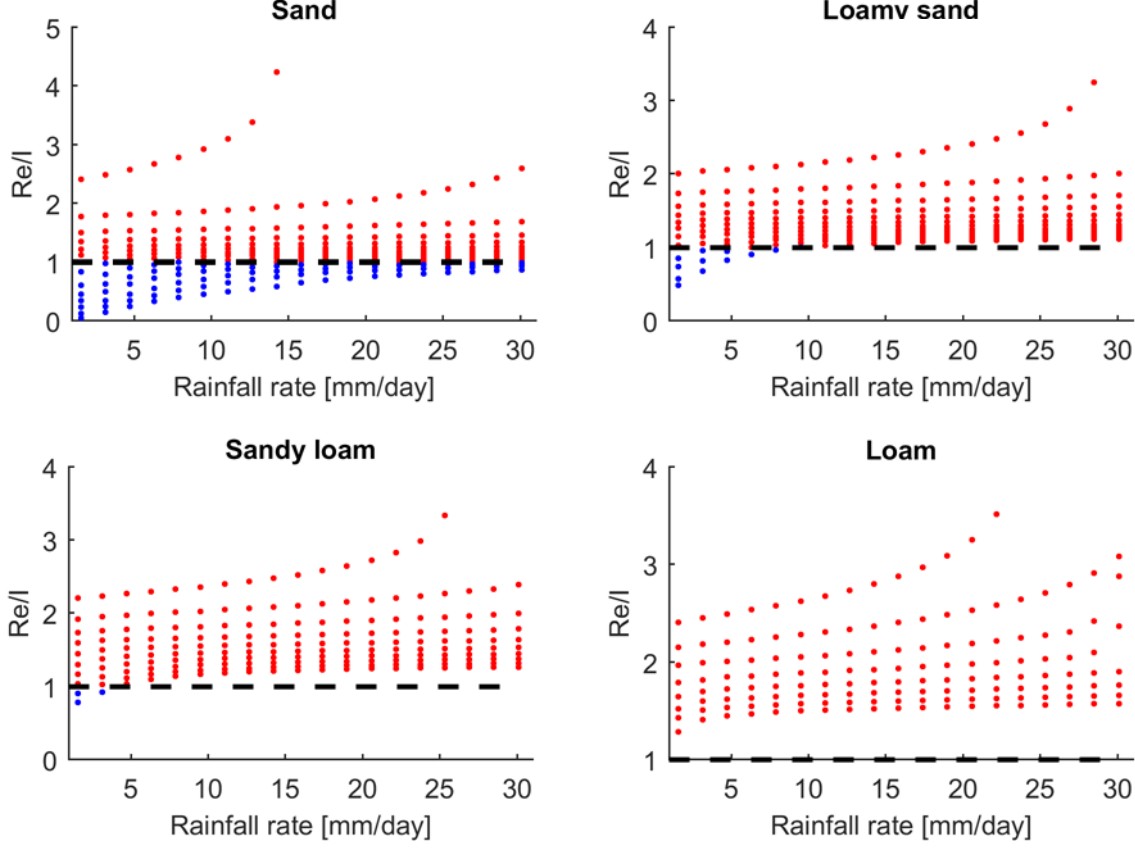

5    **Figure 4. Variations of the ratio of recharge/infiltration with rainfall and initial depth to water table in different soil types simulated by MIKESHE. The red dots are $\frac{Re}{I} \geq 1$ and the blue dots are $\frac{Re}{I} < 1$.**





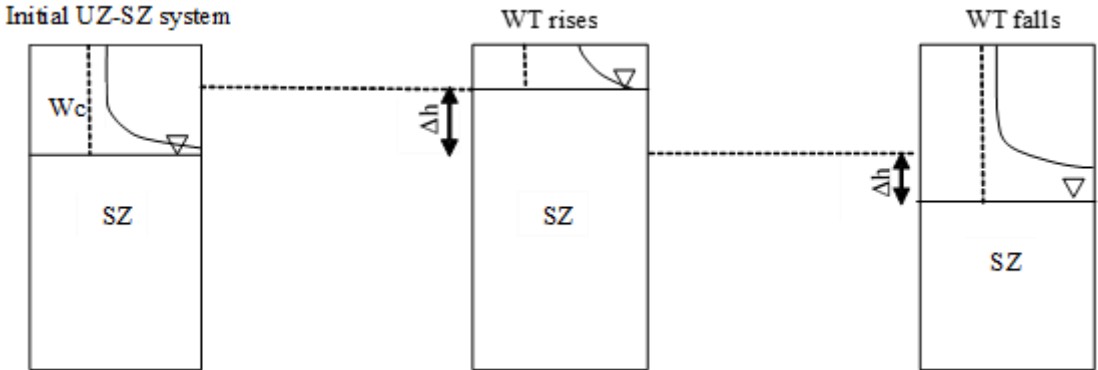

**Figure 5. Conceptualization of the interaction between UZ-SZ in MOBIDIC-MODFLOW in rising and falling water tables.**

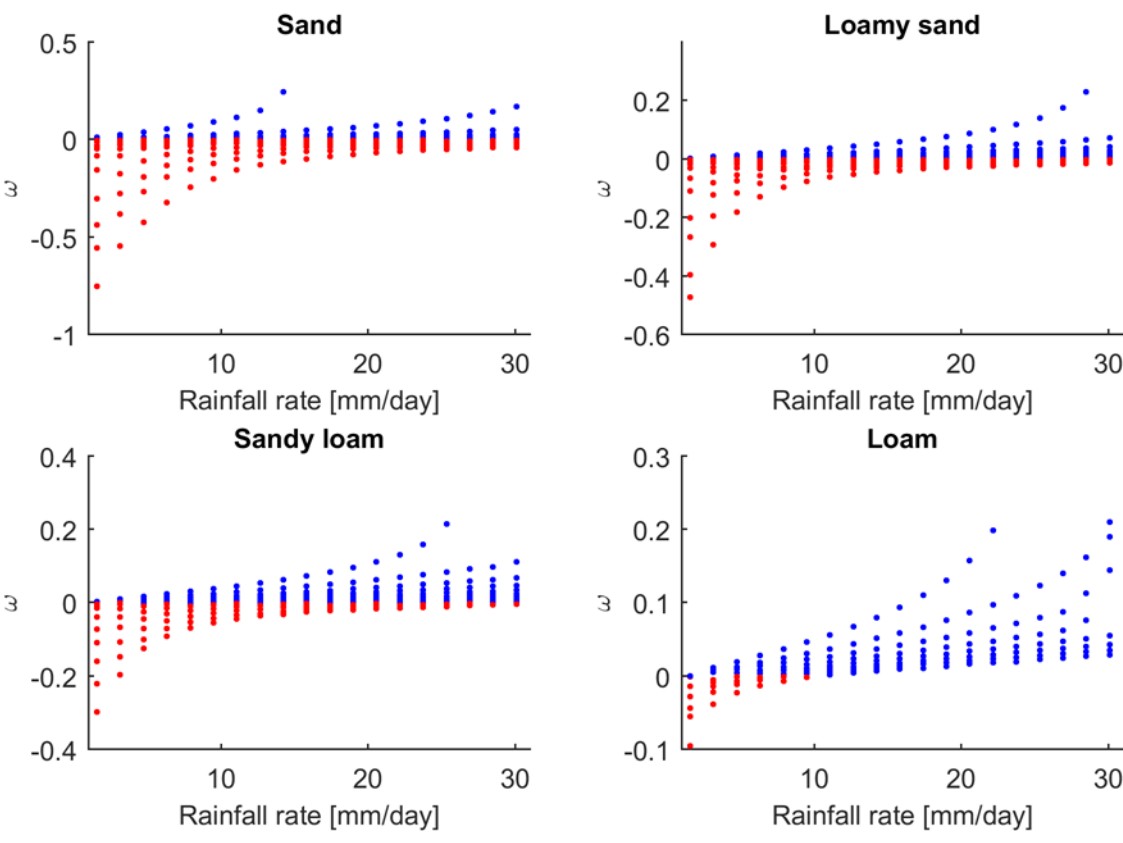

**Figure 6. Variations in $\omega$ against rainfall intensity for different initial water table level and soil types.**





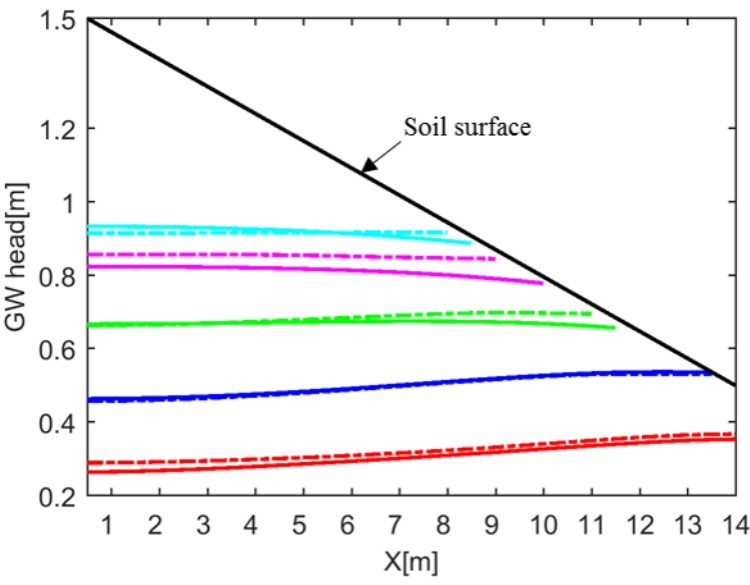

**Figure 7. Simulated water table level after 4 (red), 8 (blue), 12 (green), 16 (magenta) and 20 days (sky blue) with MIKESHE (solid lines) and MOBIDIC-MODFLOW (dashed lines).**

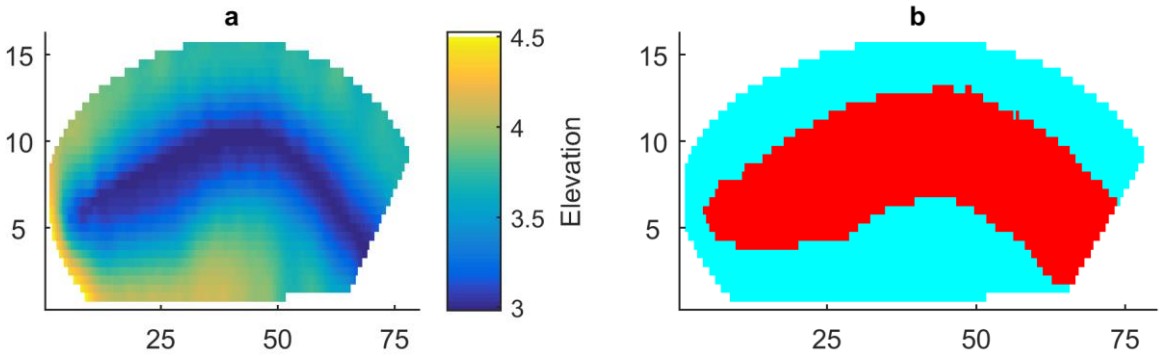

5    **Figure 8. a) The digital elevation model of Borden catchment and b) zoning map of the catchment based on depth to water table less than 1.5m (red) and greater than1.5m (blue).**


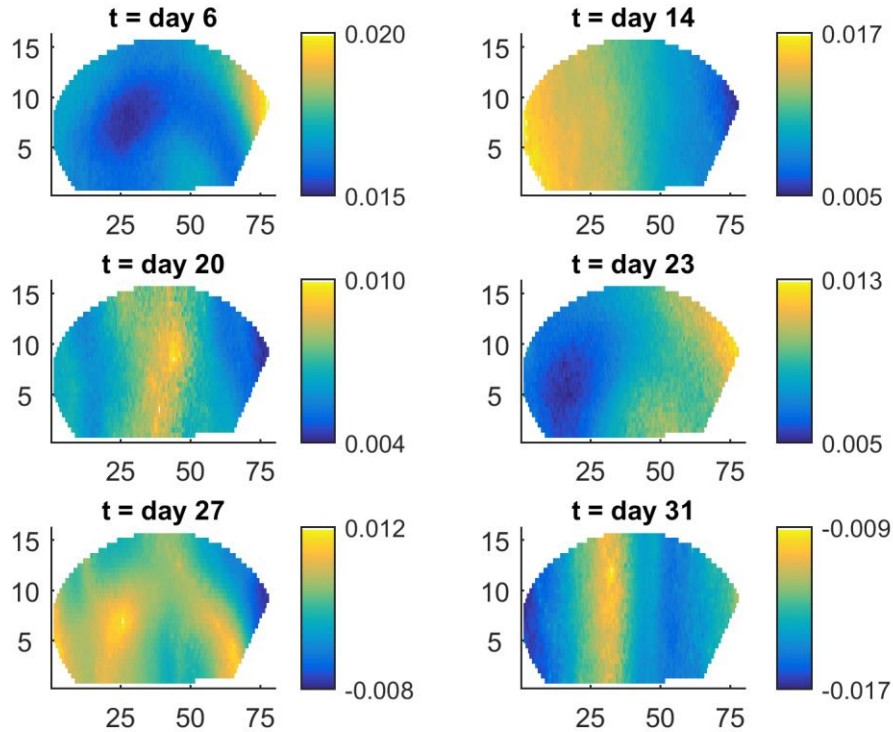

**Figure 9. The difference in simulated water tables by MIKESHE and MOBIDIC-MODFLOW ($h_{MOBIDIC} - h_{MIKESHE}$) [m] at different time steps.**

5     **Table1. Hydraulic properties of the soil types used in this study based on (Rawls et al., 1982) and simulated range of initial water table depths.**

| Parameter | Sand | Loamy sand | Sandy loam | Loam |
|:---:|:---:|:---:|:---:|:---:|
| $\theta_{sat}$ | 0.437 | 0.437 | 0.453 | 0.463 |
| $\theta_{fld}$ | 0.091 | 0.125 | 0.207 | 0.270 |
| $\theta_{res}$ | 0.02 | 0.035 | 0.041 | 0.027 |
| $K_s [{cm}/{hr}]$ | 21 | 6.11 | 2.59 | 1.32 |
| $\psi_1 [cm]$ | 15.98 | 20.58 | 30.2 | 40.12 |
| $m$ | 0.694 | 0.553 | 0.378 | 0.252 |
| Initial depth to water table | 0.3m-1.5m | 0.5m-1.5m | 0.7m-1.5m | 0.9m-1.5 |



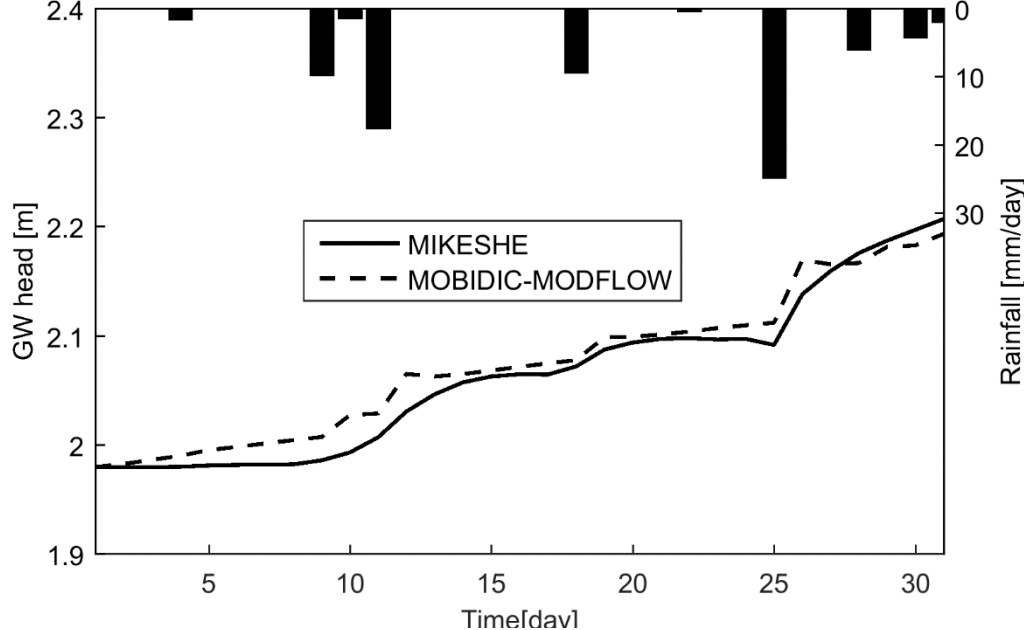

**Figure 10. Simulated water table depth by MIKESHE (solid lines) and MOBIDIC-MODFLOW (dashed lines) at the outlet grid of Borden catchment.**