# Peer review of "Modelling of shallow water table dynamics using conceptual and physically based integrated surface water-groundwater hydrologic models"

_Hydrology and Earth System Sciences, 2018_

## Referee Comment (RC1) · Anonymous Referee #1 · 6 Nov 2018

General comments

The manuscript studied a conceptual scheme of the interaction between unsaturated and saturated zones of the MOBIDIC hydrological model which is applicable to shallow water table conditions. This is an interesting topic. However, it will still need some clarification. The results and discussion section needs further improvement, compare your findings with the other author's findings. And novelty of this work should be clearly explained. I suggested major revision of this current version.

[Figure]

Specific comments

Page 3, Line 10: Please add a clear statement of what is the objective of this study at the end of the introduction part. Page 3, Line 20: The "evapotranspiration from groundwater (ðİŘÿðİŚĞðİŘžðİŚŁ)" is not clear for all the readers. Please add an explanation of this concept.

Page 4, Line 10: I feel this is not clear what are the differences between the MIKE-SHE and the MOBIDIC hydrological model. You may use a table to show the differences or what you have improved or changed.

Page 8, line 15: You may use a flowchart here to show the methods you used or the model setup processes. Page 14, line 20: I suggest the author add some paragraphs to compare this study and the previous similar studies.

Page 15, line 10: You mentioned that the efficiency of the new model is better than MIKE-SHE, but what about the uncertainty of the new model than MIKE-SHE? Because you added new parameters here.

Figure 9: What about the situation after 31 days. Also, do you have observation of the spatial distribution water tables?

Figure 10: Do you have observations of water depth to compare with the model simulations.

---

## Referee Comment (RC2) · Anonymous Referee #2 · 20 Nov 2018

The paper is well organized/ written and for further improvement, I would like to forward the following comments:

(1) In this paper, the authors are reporting as they present a new conceptual scheme of coupled MOBIDIC-MODFLOW model. But, in the paper, nothing is said about MOD-FLOW and how they link the two models. (2) In this study, the MOBIDIC-MODFLOW results were based on the output of MIKE-SHE (eg. the coefficient of groundwater recharge used by the model is based on the water table of MIKE-SHE) and the result also interpreted again by comparing with MIKE-SHE results. How much this coupled

model can stand alone without MIKE-SHE? Why not consider the evaluation of the model result by comparing the measured time series water table of the month considered in the "real" field condition? (3) Page 1 (line 21) and page 16 (lines 23-26)- It is reported that in computational efficiency (time efficiency) of the proposed approach, MIKE-SHE took 180 times longer to solve the 3D case than the MOBIDIC-MODFLOW in its application to real catchment case studies. Since MIKE SHE model simulation covers a fully integrated aspect of all important hydrology including groundwater, surface water, recharge, and evapotranspiration, how much the new coupled model is capable in computing all those hydrological processes, and is it acceptable to compare the efficiency of the two models and report theses much gap? (4) Page 2 (Line 15) - "Inconsistency in the conceptualization of the interaction between SZ and UZ" is reported in externally linked models listed. It needs a strong justification. The recently released SWAT_MODFLOW papers could not agree with this idea. (5) There is inconsistence in using the abbreviation for moisture content at saturation which is used in page 5 line 14. (6) "t" is missed in the ward water table in sentences on page10 line 8 and page 14 line 13. (7) A full stop (.) is missed in the sentence on page 13 line12.

---

## Author Comment (AC1) · 28 Nov 2018

**Response to Anonymous Reviewer #1**

The authors would like to express their sincere gratitude to the reviewer for his/her insightful comments which will surely enhance the paper. We have revised the paper based on your suggestions. Our responses (in black) to the questions (in blue) are described below.

- General comments

*''The manuscript studied a conceptual scheme of the interaction between unsaturated and saturated zones of the MOBIDIC hydrological model which is applicable to shallow water table conditions. This is an interesting topic. However, it will still need some clarification. The results and discussion section needs further improvement, compare your findings with the other author's findings, Please add a clear statement of what is the objective of this study at the end of the introduction part and novelty of this work should be clearly explained.''*

Thank you for your great suggestion. We have revised the introduction part to better explain the objective and novelty of this study:

Considering the limitations associated with the application of the MOBIDIC-MODFLOW in the very shallow water table regions (type 3), the objective of this study is to propose a series of modifications to the original conceptualization of the hydrological processes of the MOBIDIC-MODFLOW to extend its potential applicability for these cases.

To this aim, a novel methodology for revisiting the calculation of the groundwater recharge in MOBIDIC, specific yield in MODFLOW, and the interaction between the unsaturated and saturated zone in MOBIDIC-MODFLOW was developed. The developed methodology is based on the premise that the ''expected'' response of a shallow water table system is given by MIKE SHE a fully coupled surface-subsurface model taken as the reference model of this study. Using Water Table Fluctuation (WTF) method (Healy and Cook, 2002), the water table rises of a shallow water table system under different sets of rainfall intensity, soil property and depth to water table were simulated using MIKE SHE. The simulated responses were then used to reformulate the groundwater recharge of MOBIDIC based on the hydrostatic equilibrium interaction between the unsaturated and saturated zones. The accuracy of the proposed changes is first tested in a two-dimensional case

where subsurface water is simulated in a vertical plane with a constant slope. A constant rainfall rate is applied and the rise in groundwater levels is affected by groundwater recharge and by the lateral interaction between the saturated computational grids. In a second numerical experiment, the accuracy of the approach is further evaluated at the catchment scale and under unsteady rainfall where the simulated water table levels of the two models (MIKE SHE as the reference model and MOBIDIC-MODFLOW) are compared.

The comparison of the simulated water table responses of the MOBIDIC-MODFLOW against those of MIKE SHE allows us to evaluate how the externally coupled models such as MOBIDIC-MODFLOW in our study can be modified for applications in very shallow water tables.

The discussion regarding the comparison of the findings of this study with others is given in page 3 of this document.

- *Specific comments*

*1- Page 3, Line 20: The "evapotranspiration from groundwater ($\Delta S_{GW} = R + Q_u - Q_u - ET_{GW}$) is not clear for all the readers. Please add an explanation of this concept.*

Thanks for your suggestion. The description of the evapotranspiration from the groundwater was added in the revised version of the paper:
The evapotranspiration from groundwater is the direct root water uptake from the saturated zone. Unlike the deep water table conditions, the groundwater evapotranspiration can be much greater than the evapotranspiration from the unsaturated zone as discussed by (Shah et al., 2007).

*2- Page 4, Line 10: I feel this is not clear what are the differences between the MIKE-SHE and the MOBIDIC hydrological model. You may use a table to show the differences or what you have improved or changed.*

Thanks for your great suggestion. The following table describing the differences between the two models was added to the revised manuscript:

Table 1. Comparison of the subsurface flow processes in MOBIDIC-MODFLOW and MIKE SHE.

| Model/Process | Unsaturated zone | Saturated zone | UZ-SZ coupling | Applications in humid regions |
|---|---|---|---|---|
| MIKE SHE | 1D Richards | 3D finite difference | Iterative water table correction in each UZ time steps | Applicable in both deep and shallow water table regions. The dynamic variations of the specific yield in shallow water table regions are handled using the UZ-SZ coupling approach. However, the iterative process increases the computational burden of the model. |
| MOBIDIC-MODFLOW | Dual reservoir | 3D finite difference | Sequential[1] coupling | Since it uses a constant specific yield, it has limitations in modelling of the water table fluctuations of the humid regions. The simplified UZ-SZ coupling approach makes the model computationally efficient. |

[1] Sequential coupling means the solution of the water table from the previous time step is used as the boundary condition for the solution of the  MODFLOW (Guzha 2008)

*3- Page 8, line 15: You may use a flowchart here to show the methods you used or the model setup processes.*

Thank you for the suggestion. The following flowchart describing the step-by-step of the procedure was added to the revised manuscript:

[Figure]

Figure 4. Flowchart describing the step-by-step procedure of water table fluctuation method in MIKE SHE. $i_{max}$ is the maximum rainfall rate below which the infiltration excess runoff doesn't occur. $\psi_1$ is the soil bubbling pressure given in Table 2.

*4- Page 14, line 20: I suggest the author add some paragraphs to compare this study and the previous similar studies.*

The authors are very thankful for this comment. The discussion part of the manuscript was revised as to include the comparison with the previous studies:

The quick rise of the shallow water table in response to the precipitation was also observed in the experimental work of (Abdul and Gillham 1984). In their study, the water table response of a sandy soil packed in a $160 \times 120 \times 8$ cm (surface slope 12°) box with different initial water table level under a uniform rainfall rate was investigated. The objective of the experiment was to evaluate the effect of the capillary fringe on the rise in water table and streamflow generation. The results revealed that when the water table is very shallow, that is for the downhill regions of the slope in which the capillary fringe extended to the soil surface, the small amount of rainfall can result in a water table rise much greater than what would be expected by the specific yield of the soil. The uphill regions with deeper water table depth, however, showed a delayed response due to the presence of the moisture deficit in the unsaturated zone. The simulated responses using MIKE SHE and MOBIDIC-MODFLOW (Figure 8) are consistent with the findings of the (Abdul and Gillham 1984) attesting their capability in capturing the effect of capillary fringe on the water table rise. Note that the coefficient of $\omega$ in MOBIDIC-MODFLOW (Equation 19) changes as the water table level rise/falls and therefore, each computation grid in Figure 8 has a different value for $\omega$ at each time step.

The quantitative comparison of the simulated water table rises of MIKE SHE and MOBIDIC-MODFLOW against the observations of (Abdul and Gillham 1984) is not possible since the soil types (sandy soil in their experiment and sandy soil for which the coefficient of $\omega$ in Equation 19 (please refer to the Table 2 and Figure 7).

The Borden catchment has also been in experimental (Abdul and Gillham 1989) and modelling studies (VanderKwaak 1999; Jones et al. 2006). However, the exact comparison of the simulation results of this work with the aforementioned studies is not possible since different soil properties are different.

*5- You mentioned that the efficiency of the new model is better than MIKE-SHE, but what about the uncertainty of the new model than MIKE-SHE? Because you added new parameters here.*

Thank you for raising this very important issue. The modified MOBIDIC-MODFLOW doesn't have any additional parameters, thereby the parameter uncertainties of the model remain unchanged. Comparison of the original and modified calculation of the groundwater recharge in MOBIDIC (Equations 12 and 19) shows that the two equations have only one calibration parameter ($\gamma$ in Equation 12 and $\omega$ in Equation 19).

The proposed modifications, however, eliminates the specific yield from the calibration parameters of MODFLOW using the water table dependent expression given in Equation 18. The required soil hydraulic parameters for application of the Equation 18 is derived based on (Rawls and Brakensiek 1989) soil database.

Therefore, the modified MOBIDIC-MODFLOW has one less calibration parameter (specific yield) compared to the original structure of the MOBIDIC-MODFLOW.

*6- Figure 9: What about the situation after 31 days. Also, do you have observation of the spatial distribution water tables?*

Thank you for the question. The comparison of the water table rises of the two models shows the water table is rising during the 31 days of simulation (see Figure 11) as they would eventually reach to the soil surface since. This is because the evapotranspiration process was not included in neither of the models. Note that we ran the models for a month since the simulations with MIKE SHE was already very time consuming (about 30 hours).

Unfortunately, we did not have any observations of the water table level to evaluate the accuracy of the predictions with the two models. In fact, the idea of this work was to propose changes in the structure of the MOBIDIC-MODFLOW to become as an alternative tool to the physically based model (such as MIKE SHE in our study) for shallow water table applications.

*7- Figure 10: Do you have observations of water depth to compare with the model simulations?*

Thanks for your question. We did not have any observations of the water depths to evaluate the accuracy of the predictions with the models. In fact, we used MIKE SHE as the reference model to evaluate the accuracy of the

predictions with MOBIDIC-MODFLOW. Also, the generated saturated excess runoff was removed from the soil surface since the surface water routing process was not considered in this work.

References

Abdul, A. S., and R. W. Gillham. 1984. "Laboratory Studies of the Effects of the Capillary Fringe on Streamflow Generation." *Water Resources Research* 20 (6): 691–98. https://doi.org/10.1029/WR020i006p00691.

Abdul, A. S., and R. W. Gillham. 1989. "Field Studies of the Effects of the Capillary Fringe on Streamflow Generation." *Journal of Hydrology* 112 (1): 1–18. https://doi.org/10.1016/0022-1694(89)90177-7.

Guzha, A. 2008. "Integrating Surface and Sub Surface Flow Models of Different Spatial and Temporal Scales Using Potential Coupling Interfaces." http://digitalcommons.usu.edu/etd/50.

Healy, Richard W., and Peter G. Cook. 2002. "Using Groundwater Levels to Estimate Recharge." *Hydrogeology Journal* 10 (1): 91–109. https://doi.org/10.1007/s10040-001-0178-0.

Jones, J. P., E. A. Sudicky, A. E. Brookfield, and Y.-J. Park. 2006. "An Assessment of the Tracer-Based Approach to Quantifying Groundwater Contributions to Streamflow." *Water Resources Research* 42 (2): W02407. https://doi.org/10.1029/2005WR004130.

Rawls, W. J., and D. L. Brakensiek. 1989. "Estimation of Soil Water Retention and Hydraulic Properties." In *Unsaturated Flow in Hydrologic Modeling*, edited by H. J. Morel-Seytoux, 275–300. NATO ASI Series 275. Springer Netherlands. http://link.springer.com/chapter/10.1007/978-94-009-2352-2_10.

Shah, Nirjhar, Mahmood Nachabe, and Mark Ross. 2007. "Extinction Depth and Evapotranspiration from Ground Water under Selected Land Covers." *Ground Water* 45 (3): 329–38. https://doi.org/10.1111/j.1745-6584.2007.00302.x.

VanderKwaak, Joel E. 1999. "Numerical Simulation of Flow and Chemical Transport in Integrated Surface-Subsurface Hydrologic Systems." Canada: University of Waterloo. https://uwspace.uwaterloo.ca/handle/10012/412.

---

## Author Comment (AC2) · 29 Nov 2018

**Response to Anonymous Reviewer #2**

The authors would like to express their deepest gratitude to the reviewer for his/her insightful comments which will surely enhance the paper. We have revised the paper based on your suggestions. Our responses (in black) to the questions (in red) are given below.

*1- In this paper, the authors are reporting as they present a new conceptual scheme of coupled MOBIDIC-MODFLOW model. But, in the paper, nothing is said about MODFLOW and how they link the two models.*

Thank you for your excellent suggestion. The coupling process of MOBIDIC and MODFLOW will be included in the revised version of the manuscript:

The MOBIDIC and MODFLOW are coupled using the sequential coupling approach discussed in (Guzha 2008). At each time step, the spatial distribution of the groundwater head determined in MODFLOW from the previous time step is transferred into MOBIDIC for calculation of the groundwater recharge in the concurrent time step (equation 12). The calculated groundwater recharge will then be used as the upper boundary condition for the calculation process in MODFLOW for the next time step. This process continues until the last time step.

*2- In this study, the MOBIDIC-MODFLOW results were based on the output of MIKE-SHE (e.g. the coefficient of groundwater recharge used by the model is based on the water table of MIKE-SHE) and the result also interpreted again by comparing with MIKE-SHE results. How much this coupled model can stand alone without MIKE-SHE? Why not consider the evaluation of the model result by comparing the measured time series water table of the month considered in the "real" field condition?*

Thanks for raising this very important issue. Although the proposed modifications in the model were evaluated against MIKE SHE as the reference model, the model itself can be used independently. The calibration of the coefficient of the groundwater recharge based on the simulation results of MIKE SHE enabled us to propose an alternative model (MOBIDIC-MODFLOW) which is computationally and parametrically simpler. This is important as the developed model is aimed to be applicable at watershed scale simulations where the computational and parametrical efficiency of the model is of great concern.

The comparison of the simulation results against MIKE SHE was made to investigate how the simplifications of in MOBIDIC-MODFLOW can affect the simulated water tables if it applies at the watershed scale. Unfortunately, we didn't have any observations of the water table to test the fidelity of the proposed modifications in MOBIDIC-MODFLOW. Consequently, the simulated water table levels of MIKE SHE was considered as the ''expected'' response of the catchment in absence of the observations.

*3- Page 1 (line 21) and page 16 (lines 23-26)- It is reported that in computational efficiency (time efficiency) of the proposed approach, MIKE-SHE took 180 times longer to solve the 3D case than the MOBIDIC-MODFLOW in its application to real catchment case studies. Since MIKE SHE model simulation covers a fully integrated aspect of all important hydrology including groundwater, surface water, recharge, and evapotranspiration, how much the new coupled model is capable in computing all those hydrological processes, and is it acceptable to compare the efficiency of the two models and report theses much gap?*

We are very thankful for this insightful question. Same as MIKE SHE, the MOBIDIC-MODFLOW is capable of simulating all aspects of the hydrologic cycle including the groundwater flow, recharge, evapotranspiration, overland, and channel flow. However, the formulations of the hydrological processes in the two models are different. For example, the overland flow and channel flow in MIKE SHE are described using the Saint-Venant equations, whereas in MOBIDIC-MODFLOW these are based on steepest descent and linear reservoir approach.

In terms of subsurface (unsaturated and saturated zone) flow, which is the subject of this paper, the differences are in the conceptualization of the unsaturated zone and its coupling process with the saturated zone. In MIKE SHE the unsaturated zone is extended from soil surface to the water table and it is described using the Richards equation. In MOBIDIC-MODFLOW with the introduced modifications, the unsaturated zone is also extended from soil surface to the water table, however, it is described using the dual reservoir approach. Such conceptual formulation of the unsaturated zone eliminates the fine spatial and temporal resolutions required in the Richards equation, resulting its computational efficiency.

Regarding the unsaturated-saturated coupling procedure, the two models have some differences. In MIKE SHE, the water table level is iteratively corrected within each unsaturated time step which is not the case in sequential coupling

approach implemented in MOBIDIC-MODFLOW (please refer to the question 1 for the detailed description of the method).

Another difference between the two models is in the formulation of the evapotranspiration process. The (Kristensen and Jensen 1975) approach in MIKESHE calculates the moisture extraction for each calculation node in the unsaturated zone. However, in MOBIDIC-MODFLOW, the capillary and gravity reservoirs are not vertically discretized and evapotranspiration loss occurs from the capillary reservoir.

Such differences in the conceptualization of the evapotranspiration process yield a different number of calibration parameters for the description of the evapotranspiration process in the two models. The (Kristensen and Jensen 1975) model have four parameters, however, the magnitude of the evapotranspiration rate in MOBIDIC-MODFLOW is controlled with only one parameter. This is an important advantageous of MOBIDIC-MODFLOW since a low number of parameters makes the calibration process more efficient and reduces the risk of equifinality issue (Beven 2001).

Note that the two models have similar formulation and solution approach for the saturated zone (the Preconditioned Conjugate Solver (PCG) solver in saturated flow module of MIKE SHE is identical to the one used in MODFLOW).

Therefore, similar to MIKE SHE, the MOBIDIC-MODFLOW covers all aspects of the hydrological process, but with different formulations. The comparison of the computational efficiency of the two models enabled us to investigate how much the simplification of the hydrological processes especially in the unsaturated zone can improve the computational efficiency of the model. As it was mentioned in question 2, the computational efficiency is an important factor for watershed scale application of the integrated model.

*4- Page 2 (Line 15) "Inconsistency in the conceptualization of the interaction between SZ and UZ" is reported in externally linked models listed. It needs a strong justification. The recently released SWAT-MODFLOW papers could not agree with this idea.*

The main problem regarding the application of the externally coupled models in shallow water table cases is the assumption of a constant specific yield. The specific yield decreases nonlinearly as the water table rises. Therefore, the rises in the water table would be much greater than what would be expected using a constant specific yield as discussed in (Abdul and Gillham 1989). Such issue hasn't been discussed in the publications of the SWAT-MODFLOW (Bailey et al. 2016; Guzman et al. 2015; Chung et al. 2010) or

TOPMODEL-MODFLOW (Guzha and Hardy 2010). With modifications in MOBIDIC-MODFLOW, we aimed to address this issue and extend the applications of the externally coupled models in shallow water table cases.

*5- There is inconsistence in using the abbreviation for moisture content at saturation which is used in page 5 line 14.*

We are very thankful for your careful reading of the manuscript. It was corrected in the revised version of the manuscript.

*6- "t" is missed in the ward water table in sentences on page10 line 8 and page 14 line 13.*

Thank you for your careful reading of the manuscript. It was corrected in the revised version of the paper.

*7- A full stop (.) is missed in the sentence on page 13 line12.*

Thank you for your careful reading of the manuscript. A full stop was added to the sentence.

**References**

Abdul, A. S., and R. W. Gillham. 1989. "Field Studies of the Effects of the Capillary Fringe on Streamflow Generation." *Journal of Hydrology* 112 (1): 1–18. https://doi.org/10.1016/0022-1694(89)90177-7.

Bailey, Ryan T., Tyler C. Wible, Mazdak Arabi, Rosemary M. Records, and Jeffrey Ditty. 2016. "Assessing Regional-Scale Spatio-Temporal Patterns of Groundwater–Surface Water Interactions Using a Coupled SWAT-MODFLOW Model." *Hydrological Processes* 30 (23): 4420–33. https://doi.org/10.1002/hyp.10933.

Beven, K. J. 2001. "How Far Can We Go in Distributed Hydrological Modelling?" *Hydrol. Earth Syst. Sci.* 5 (1): 1–12. https://doi.org/10.5194/hess-5-1-2001.

Chenjerayi Guzha, Alphonce, and Thomas Byron Hardy. 2010. "Simulating Streamflow and Water Table Depth with a Coupled Hydrological Model." *Water Science and Engineering* 3 (3): 241–56. https://doi.org/10.3882/j.issn.1674-2370.2010.03.001.

Chung, Il-Moon, Nam-Won Kim, Jeongwoo Lee, and Marios Sophocleous. 2010. "Assessing Distributed Groundwater Recharge Rate Using Integrated Surface Water-Groundwater Modelling: Application to Mihocheon Watershed, South Korea." *Hydrogeology Journal* 18 (5): 1253–64. https://doi.org/10.1007/s10040-010-0593-1.

Guzha, A. 2008. "Integrating Surface and Sub Surface Flow Models of Different Spatial and Temporal Scales Using Potential Coupling Interfaces." http://digitalcommons.usu.edu/etd/50.

Guzman, J. A., D. N. Moriasi, P. H. Gowda, J. L. Steiner, P. J. Starks, J. G. Arnold, and R. Srinivasan. 2015. "A Model Integration Framework for Linking SWAT and MODFLOW." *Environmental Modelling & Software* 73 (November): 103–16. https://doi.org/10.1016/j.envsoft.2015.08.011.

Kristensen, K. J., and S. E. Jensen. 1975. "A MODEL FOR ESTIMATING ACTUAL EVAPOTRANSPIRATION FROM POTENTIAL EVAPOTRANSPIRATION." *Hydrology Research* 6 (3): 170.

---

## Author Response (AR1)

Dear Editor and reviewers,

We would like to express our gratitude for your valuable comments/suggestions. The manuscript has been thoroughly revised according to your comments/suggestions. The summary of modifications follows:

1- The introduction has revised to better explain the objective and novelties of the studies asked by reviewer #1.

2- The section 3.2.2 (description of the saturated flow module of MOBIDIC) was changed as to include the coupling process of MOBIDIC and MODFLOW pointed by reviewer #2.

3- A new table (table 1) was added to the manuscript in which the unsaturated-saturated flow schemes of the two models (MIKE SHE as the reference model and MOBIDIC-MODFLOW) are compared. This was done, to better highlight the necessity of the modifications introduced in this study for application of MOBIDIC-MODFLOW in shallow water table regions asked by reviewer #1.

4- A new Figure (Figure 4) was added in section 4 (water table fluctuation method) to explain the step-by-step of the process asked by reviewer #1.

5- The discussion section was completely revised to include the comparison with the findings of other studies, as well as detailing the parameter uncertainty of the new unsaturated-saturated flow scheme of the model asked by reviewers # 1 and 2.

We hope that the revisions met your expectations and make the manuscript suitable for publication in HESS.

Best regards,

Mohammad Bizhanimanzar, on behalf of authors.

[revised manuscript text omitted]

---

## Referee Report (RR1)

The author modified the manuscript according to the comments of the reviewers. However, there are still some issues need to be addressed.

Specific comments:
1. I feel that the case study in this research seems a little simple. I still suggest to add more case studies, especially some simulations at catchment scale and comparison with observations.
2. I suggests that the author could add more descriptions about the computation efficiency of the new model and MIKE SHE to show the advantage of the new model.
3. In Figure 10, from day 6 to day 31, the difference in simulated water tables by MIKE SHE and MOBIDIC-MODFLOW seems not significantly decreased. However, the spatial pattern was changed a lot. This needs further explanation.
4. Figure 11, simulated water table depth by MIKE SHE are smooth but the water table depth by MOBIDIC-MODFLOW are not smooth. Please explain it.
5. In equation 19, I feel that the meaning of parameter $\omega$ needs to be further explained. What are the results of the calibration of $\omega$?

---

## Author Response (AR3)

Dear Editor,

We would like to thank you for reviewing our paper and sharing the reviewer's comments with us. A point-by-point answer to each of the comments/questions is given below.

*1- I feel that the case study in this research seems a little simple. I still suggest to add more case studies, especially some simulations at the catchment scale and comparison with observations.*

Testing the models in different case studies is a valuable suggestion. However, we do not have such data. Comparison of the simulated responses of our MOBIDIC-MODFLOW against MIKE SHE was made assuming the latter's simulated response as the "expected" – or "benchmark" response of the catchment in absence of observations. Such comparison allows to effectively demonstrate how close the revised model can mimic the "expected" response of the system and what are the reasons for the discrepancies. We selected the Borden catchment case as it is one of the widely used catchment scale case study in many recent integrated surface water-groundwater hydrologic model development e.g., [1], [2], and [3] which were cited in the manuscript (please refer to the section 6.2.1).

*2- I suggest that the author could add more descriptions about the computation efficiency of the new model and MIKE SHE to show the advantage of the new model.*

[Please refer to lines 4-14 page 17 of the revised manuscript] With regards to the similarities in the formulation and numerical solution technique of saturated flow in MIKE SHE and MOBIDIC-MODFLOW, the computational efficiency of our updated MOBIDIC-MODFLOW model is related to the conceptualization of the unsaturated flow and its interaction with the saturated zone. In MIKE SHE, the numerical solution of the Richards' equation imposes very short time steps, i.e. in order of seconds and the soil profile is discretized into computational nodes. In addition, the iterative unsaturated-saturated coupling method in each unsaturated zone time step can remarkably increase the execution time of the system.

In MOBIDIC-MODFLOW, however, there is only one calculation node in the unsaturated zone of the soil profile (dual pore reservoir approach of MOBIDIC) and a daily time step is applied for both unsaturated and saturated flow calculations. The unsaturated (MOBIDIC) and saturated (MODFLOW) zones are coupled through a sequential coupling technique (described in section 3.2.2) and our coupling scheme (which includes a calibrated groundwater recharge equation) presented in this study keep the high computational efficiency of the coupled model.

[please refer to lines 10-14 page 16 of the revised manuscript] It is worth mentioning that the number of calibration parameters in the revised MOBIDIC-MODFLOW remains unchanged as equations 12 (calculation of groundwater recharge in original MOBIDIC-MODFLOW) and 19 (revised groundwater recharge) have one calibration parameter (respectively $\gamma$ and $\omega$).

*3- In Figure 10, from day 6 to day 31, the difference in simulated water tables by MIKE SHE and MOBIDIC-MODFLOW seems not significantly decreased. However, the spatial pattern was changed a lot. This needs further explanation.*

Please refer to the discussion given in section 6.2.2 page 15 lines 2-11. The differences in the spatial pattern of figure 10 are related to the lateral saturated flow between zones 1 and 2 during the course of simulation. Zone 1 (red in Figure 9b) has an initial depth to water table less than 1.5 m and is simulated using the revised unsaturated-saturated scheme presented in this study (section 5). However, water table level simulations for grids with depth to water table larger than 1.5 m i.e., zone 2 (blue

in Figure 9b) is performed using the original UZ-SZ scheme (section 3.2.3). [Added in the section 6.2.2] In Figure 10 at day 6, MOBIDIC-MODFLOW simulates higher water table levels compared to MIKE SHE which is because of the higher groundwater recharge received by the cells in zone 2 and subsequent lateral saturated flow from zone 2 to zone 1. Note that in the original UZ-SZ scheme of MOBIDIC, if the groundwater level is below the modelled soil layer (as in zone 2), the

5 groundwater recharge is a linear function of moisture state of gravity reservoir (see equation 12). However, in MIKE SHE, the groundwater recharge is calculated from the solution of Richards equation considering the water table level as the lower boundary condition. Therefore, the magnitude of groundwater recharge simulated by MIKE SHE for zone 2 is smaller than MOBIDIC-MODFLOW. The spatial pattern of water table depth at day 14 and following events changes as the direction of saturated lateral flow between the grids changes.

10 *4- Figure 11, simulated water table depth by MIKE SHE are smooth but the water table depth by MOBIDIC-MODFLOW is not smooth. Please explain it.*

[Added in section 6.2.2 page 15 lines 12-20] In Figure 11, the water table changes (sharp in MIKE SHE and smooth in MOBIDIC-MODFLOW) are related to the calculation of groundwater recharge in the two models. Equation 19 (revised groundwater recharge) shows that in the revised version of MOBIDIC-MODFLOW, the groundwater recharge is zero during

15 the days without infiltration. This means that the response of water table to the fallen precipitation is rapid and unsaturated moisture profile quickly reaches its equilibrium state as discussed in section 5. Therefore, on rainy days, the MOBIDIC-MODFLOW's water table quickly rises as opposed to that of MIKE SHE, in which the groundwater recharge is determined from the solution of Richards' equation and the profile may take a longer time before reaching its equilibrium state. Note that the slight increase in water table level of MOBIDIC-MODFLOW between days 14 to 18 (no rainfall period) is due to the

20 incoming lateral saturated flow from adjacent cells.

*5- In equation 19, I feel that the meaning of parameter $\omega$ needs to be further explained. What are the results of the calibration of $\omega$?*

Please refer to section 5.1 for detailed interpretation of the parameter $\omega$ in equation 19. The parameter of $\omega$ is a function of rainfall rate, soil type, and depth to water table. When it takes

25 a positive value, the groundwater recharge becomes higher than the infiltration rate and the opposite occurs when it is negative. This parameter was calibrated based on water table rise of MIKE SHE for different combinations of soil type, rainfall intensity and depth to water table as shown in Figure 7. From figure 7, it can be seen that for a given rainfall rate and depth to water table, loamy soils take more positive values than sandy soils, which means higher water table rise would be expected with this soil type. The higher water table rise is associated with very small specific yield for this soil type which causes the rises to be

30 much smaller than expected by the equilibrium constant specific yield. Such methodology enabled us to implement complex behavior of water table responses of different soil types to the rainfall intensity using a conceptual scheme that can be implemented in externally coupled surface water-groundwater models such as MOBIDIC-MODFLOW for applications in humid regions.

**References**

[number 1 and 3 were already in the manuscript and number 2 was added in the revised version od manuscript]

[revised manuscript text omitted]